# 2,2-Bis(4-Hydroxyphenyl)-1-Propanol—A Persistent Product of Bisphenol A Bio-Oxidation in Fortified Environmental Water, as Identified by HPLC/UV/ESI-MS

**DOI:** 10.3390/toxics9030049

**Published:** 2021-03-05

**Authors:** Małgorzata Drzewiecka, Monika Beszterda, Magdalena Frańska, Rafał Frański

**Affiliations:** 1Faculty of Chemistry Adam Mickiewicz University, Uniwersytetu Poznańskiego 8, 61-614 Poznań, Poland; drzewiecka.m@onet.pl; 2Department of Food Biochemistry and Analysis, Poznań University of Life Sciences, Mazowiecka 48, 60-623 Poznań, Poland; monika.beszterda@gmail.com; 3Institute of Chemistry and Technical Electrochemistry, Poznań University of Technology, Berdychowo 4, 60-965 Poznań, Poland; magdalena.franska@put.poznan.pl

**Keywords:** bisphenol A, bio-oxidation, 2,2-bis(4-hydroxyphenyl)-1-propanol, environmental water, mass spectrometry

## Abstract

Biodegradation of bisphenol A in the environmental waters (lake, river, and sea) has been studied on the base of fortification of the samples taken and the biodegradation products have been analyzed using HPLC/UV/ESI-MS. Analysis of the characteristic fragmentation patterns of [M-H]^−^ ions permitted unambiguous identification of the biodegradation products as 2,2-bis(4-hydroxyphenyl)-1-propanol or as *p*-hydroxyacetophenone, depending on the type of surface water source. The formation of 2,2-bis(4-hydroxyphenyl)-1-propanol was much more common than that of *p*-hydroxyacetophenone. Moreover, 2,2-Bis(4-hydroxyphenyl)-1-propanol has not been further biodegraded, in contrast to the *p*-hydroxyacetophenone, which was further mineralized. It has been proved, for the first time, that 2,2-bis(4-hydroxyphenyl)-1-propanol can be regarded as persistent product of bisphenol A biodegradation in the fortified environmental waters.

## 1. Introduction

Bisphenol A (BPA, 2,2-bis(4-hydroxyphenyl)propane) is a well-known precursor of plastics, mainly epoxy resins and polycarbonates, and it is one of the endocrine-disrupting chemicals produced in large volumes worldwide. The wide use of the products based on BPA implies a high possibility of environmental contamination by BPA, mainly of lakes and rivers [1]. It has prompted a vast number of studies devoted to the biodegradation of bisphenol A as recently discussed in detail in review papers [2,3,4,5].

The biodegradation processes observed in laboratory conditions reflects the processes occurring in the natural environment. However, it is well known that even a small change in the conditions can substantially affect biological processes. Therefore, it should be ascertained if the processes observed in the laboratory are really the same as those occurring in the environment. In this study, we decided to evaluate the biodegradation of BPA in the environmental waters (lake, river, and sea) by fortification with BPA in water-in laboratory conditions. To the best of our knowledge, such simple (or even trivial) experiment has not previously been performed.

In many studies, the proposed first step of BPA biodegradation pathway is BPA oxidation (bio-oxidation) and formation of two isomers shown in Scheme 1, namely, 2,2-bis(4-hydroxyphenyl)-1-propanol and/or 1,2-bis(4-hydroxyphenyl)-2-propanol (further referred to as Product 1 and Product 2, respectively, Scheme 1) [6,7,8,9,10,11,12,13,14].

This report provides the evidence, obtained by HPLC/UV/ESI-MS, for formation of 2,2-bis(4-hydroxyphenyl)-1-propanol (Product 1) as a persistent product of BPA biodegradation in the environmental waters. The second BPA biodegradation product, although, less common, was *para*-hydroxyacetophenone (*p*-HAP, Scheme 2), which immediately underwent mineralization (decomposition to the inorganic compounds, mainly CO_2_ and H_2_O).

From the point of view of the chemistry of the biodegradation process, the structures of biodegradation products are the most important. It is obvious that the room conditions are different from the environmental ones and may be regarded only as an approximation of the latter. On the other hand, it is likely that the biodegradation of BPA in the environment occurs in a similar way, since in our experiments, we did not change the composition of environmental water.

## 2. Materials and Methods

Bisphenol A (≥99%) and *p*-HAP (99%) standards were obtained from Sigma-Aldrich (Poznań, Poland) and were used without purification. The water samples used for the tests were taken in Poland from the Warta River (the main river in the region; sampling in Poznań), from the Baltic Sea (the nearest sea; sampling in Rewal), and from 10 lakes in the middle of the Wielkopolska region (Appendix A): The water samples were collected in spring and summer, respectively. About 5 L of each water was collected by grab sampling into glass bottles at the depth of about 1 m by a sampler, leaving adequate head space for aeration and kept at a temperature of +4 °C for no longer than 72 h before the biodegradation test. The pH of all water samples was around pH 7. The selected data obtained for Niepruszewo Lake, Lusowo Lake, and the Warta River are shown as representative examples. 

The degradation tests were performed in 200 mL bottles filled with environmental water samples to which 2 mg of BPA (10 mg/L) were added. Similar concentrations were used by other authors for biodegradation test performance. The bottles were kept at room temperature (20–25 °C), opened, and exposed to the day sunlight. Every fifth day, 1 mL was collected from each bottle and 0.5 mL of methanol was added to it, and the mixtures were placed in the fridge, to cease the biodegradation process. The samples were collected over 3 months and then subjected to HPLC/UV/ESI-MS analysis.

The HPLC/UV/ESI-MS analyses were made on a Waters model 2690 HPLC pump (Milford, MA, USA), Waters 996 Photodiode Array Detector and Waters/Micromass ZQ2000 mass spectrometer (single quadrupole type instrument equipped with electrospray ion source, Z-spray, Manchester, UK). The HPLC/UV and HPLC/MS are the two most commonly used methods used for analysis of bisphenol A and its metabolites [15,16,17,18]. The software used was MassLynx V3.5 (Manchester, UK). Using an autosampler, the sample solutions were injected onto the XTerra^®^ RP18 column (5 µm, 150 mm × 3 mm i.d.; Waters, Warsaw, Poland). The injection volume was 10 µL. The solutions were analyzed using linear gradient of CH_3_CN-H_2_O with a flow rate of 0.4 mL/min. Two gradients were used, namely, acidified and non-acidified. We found that acidified gradient was better for UV–VIS detection and non-acidified was better for MS detection (MS detection was performed in negative ion mode). The acidified gradient started from 0% CH_3_CN–95% H_2_O with 5% of a 10% solution of formic acid in water, reaching 95% CH_3_CN after 30 min, and the latter concentration was kept for 10 min. The non-acidified gradient started from 5% CH_3_CN–95% H_2_O, reaching 95% CH_3_CN after 30 min, and the latter concentration was kept for 10 min. Thus, the full time of HPLC/ESI-MS analysis was 40 min, however, for the sake of clarity, the chromatograms are shown for a smaller time range. As expected, HPLC/UV yielded better linearity (signal intensities vs. compound concentration, Appendix A), whereas HPLC/ESI-MS allowed identification of biodegradation products.

The UV–VIS spectra were recorded in the range of 210–600 nm. The BPA and its degradation products were monitored by absorbance at 280 nm [6,7,19,20,21,22,23,24]. For each sample, the analyses were performed three times and the calculated relative standard deviations for the peak areas obtained upon HPLC/UV analysis did not exceed 5%.

The ESI mass spectra were recorded in the *m/z* range of 70–1000, in negative ion mode. The electrospray source potentials were: capillary 3 kV, lens 0.5 kV, extractor 4 V, and cone voltage 30–80 V. It is known that cone voltage has the greatest impact on the mass spectra recorded. Increase in this parameter leads to the so-called “in-source” fragmentation/dissociation but a too low cone voltage may cause a decrease in sensitivity. The source temperature was 120 °C, and the desolvation temperature was 300 °C. Nitrogen was used as the nebulizing and desolvation gas at the flow rates of 100 and 300 L/h, respectively.

To corroborate the structures and fragmentation patterns of Product 1, we collected the eluate containing this compound (in the proper range of retention time) and then the eluate was directly infused into the Q-TOF mass spectrometer (coupling off-line of HPLC to ESI-MS/MS), as described in the Appendix A.

## 3. Results and Discussion

At first, it has to be checked if we really deal with a biodegradation process, and not with simple BPA oxidation by air. Therefore, BPA has been also added to pure water (the tap water used was purified/deionized using demineralizer). It has to be stressed that in pure water, even after 3 months, the BPA concentration was not lowered (Appendix A). When BPA was added to the environmental water from a lake or river, after several days, its concentration started to decrease. When BPA was added to the sea water, its concentration was only slightly decreased after several dozen days, no biodegradation products were detected (Appendix A).

Figure 1 shows exemplary UV chromatograms. When BPA was stored in samples of environmental water, its concentration decrease was accompanied by an increase in the concentration the compounds characterized with a retention time of 12.6 min (Figure 1), for most of the fortified environmental water samples.

Exemplary breakdown plots of chromatographic peak areas (chromatograms obtained at 280 nm) against days of biodegradation test are presented in Figure 2. The peak areas expressed in arbitrary units were converted into percentages (the largest peak is assumed as 100%).

As clearly seen from Figure 2, the rate of BPA biodegradation strongly depends on the source of environmental water taken for the test. However, in each test, the biodegradation product characterized with a retention time of 12.6 min was not further degraded.

HPLC/ESI-MS analyses have indicated that molecular mass of the biodegradation product at the retention time of 12.6 min is 244 (ion [M-H]^−^ at *m/z* 243). In order to unambiguously determine the structure of the biodegradation product, we performed the HPLC/UV/ESI-MS analysis using non-acidified gradient, since it allowed obtaining much higher signals to noise ratio of ions [M-H]^−^ and product ions. Figure 3 shows the exemplary chromatograms obtained using non-acidified gradient.

The HPLC/ESI-MS analysis performed using high cone voltage allowed obtaining a fragmentation pathway and, as a consequence, allowed determination of the structure of Product 1. Namely, at a high cone voltage, the abundant fragment ion at *m*/*z* 211 was formed as shown in Figure 4 and in the Appendix A.

The product ion at *m/z* 211 was formed by the loss of methanol from [M-H]^−^ ion, and formation of this fragment ion cannot be expected for Product 2 (Scheme 1). In view of the above, the fragment ion at *m/z* 211 can be treated as a diagnostic ion proving that we deal with Product 1. The product ion spectra (collision-induced dissociation-MS/MS) obtained using a Q-TOF mass spectrometer confirmed the fragmentation pattern of Product 1 (Appendix A). On the other hand, our results are different from those described by Sasaki et al. who have observed identical fragmentation patterns for Product 1 and 2, regardless of minor differences in relative ion abundances [10].

Theoretically, it is possible that we deal with bio-oxidation in aromatic ring and formation of 2-(4-hydroxyphenyl)-2-(3′,4′-dihydroxyphenyl)propane, however, the fragmentation pattern of this compound is substantially different than that observed in our work [22,25,26].

BPA is estrogenic and antiandrogenic compound, whereas 2,2-bis(4-hydroxy-phenyl)-1-propanol is weekly estrogenic, non-antiandrogenic, and less toxic than parent compound in in vitro and in vivo reporter assays [27]. As indicated by Suzuki et al. 2,2-bis(4-hydroxyphenyl)-1-propanol and 2-(4-hydroxyphenyl)-2-(3′,4′-dihydroxyphenyl) propane displaced the 17β-estradiol bound to the ERa (estrogen receptor α) in a competitive manner, however, the competitive potency of these compounds was 50 times less than that of diethylstilbesterol [11]. Moreover, in human cultured MCF-7 breast cancer cells, 2,2-bis(4-hydroxyphenyl)-1-propanol did not cause proliferation. Therefore, bio-oxidation of BPA into 2,2-bis(4-hydroxyphenyl)-1-propanol seems to be justified.

Only for two of the biodegradation tests, a biodegradation pathway different than that described above has been observed. Namely, the biodegradation product characterized by a retention time of 10.3 min was observed as shown in Figure 5.

The *m/z* 135 of [M-H]^−^ ion and fragment ions at *m/z* 120, 93, 92 (Appendix A) indicate that it is *para*-hydroxyacetophenone (*p*-HAP, piceol) [28,29]. The HPLC/UV/ESI-MS analysis of *p*-HAP standard fully confirmed the structure of the biodegradation product characterized by the retention time 10.3 min. Figure 6 shows the breakdown plots of chromatographic peak areas against days of biodegradation test, for the test in which *p*-HAP was formed.

In contrast to Product 1, *p*-HAP is not a persistent biodegradation product, since it is also biodegraded, as shown in Figure 6. After the initial concentration increase, its concentration decrease was observed, however, its further metabolites were not detected.

It has to be stressed that *p*-HAP can be formed from 1,2-bis(4-hydroxyphenyl)-2-propanol (Product 2, Scheme 1) as suggested in a number of papers [4,5,7,8,9,11,12,13,14,21,30,31,32]. Therefore, bio-oxidation of BPA into Product 2 in the environmental water is possible, however, Product 2 is immediately converted into *p*-HAP (most probably through 4,4′-dihydroxy-α-methylstilbene). Furthermore, it has been also suggested that the produced *p*-HAP is further mineralized [4,7,9,11,14,30,32], it can explain why further metabolites of *p*-HAP have not been detected. 

As indicated by Ike et al. (2002), the acute toxicity and estrogenicity of BPA can be considerably eliminated by aerobic biodegradation [33]. Among above products *p*-HAP shows much lower toxicity than BPA, lack of mutagenic activity, however, might have a weak estrogenicity understood as dose-dependent increase in β-galactosidase activity [34].

## 4. Conclusions

It has been proved that under the conditions used, two BPA biodegradation pathways in the environmental waters are possible. The first, most common, is the BPA bio-oxidation and leads to the formation of 2,2-bis(4-hydroxyphenyl)-1-propanol, which is not further biodegraded, thus 2,2-bis(4-hydroxyphenyl)-1-propanol can be regarded as a persistent BPA metabolite. The second, less common, pathway is the formation of *p*-HAP, which is further mineralized. Of course, as the target is to maintain water purity, the second pathway seems to be more desirable. In order to explain why sometimes the BPA biodegradation comprises the formation of 2,2-bis(4-hydroxyphenyl)-1-propanol (Product 1, Scheme 1) and sometimes the formation of *p*-HAP, detailed biological studies should be performed.

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
