# Peer review of "2,2-Bis(4-Hydroxyphenyl)-1-Propanol—A Persistent Product of Bisphenol A Bio-Oxidation in Fortified Environmental Water, as Identified by HPLC/UV/ESI-MS"

_toxics, 2021, doi:10.3390/toxics9030049_

Round 1

Reviewer 1 Report

Introduction

Lines 39-41: Please, explain better the experiment you performed, providing more information about the “water”, “laboratory conditions” and, in case, the involvement of microorganisms.

Line 150: since HPLC/UV/ESI-MS is the analytical method used in this research, I suggest to emphasize its importance for the environmental analysis of BPA and its metabolic products in respect to the literature.

Lines 53-55: I suggest to add if these products are more or less toxic than bisphenol A.

Results and discussion

Lines 61 and 62: pure water: do you mean “sterilized water”? or you bought pure water without microorganisms and enzymes? How do you keep the purity of the water? please, clarify and integrate also in materials and methods.

I suggest to uniform all the y-axes of the chromatograms of figure 1, so it is more evident the concentration reduction.

Line 139: is this compound volatile?

Since you referred to biodegradation processes involved in BPA in your experiments, I strongly suggest to add in the discussion some evidences reported in the literature of possible microbial candidates, which could be responsible of the formation of products observed.

Materials and methods

Please, add references to this section to support the methods you used.

Lines 167-169: “in the followings…”. Please, clarify that you refer to data of chemical analysis you performed.

Lines 171: please, uniform volume units (liters or dm3) in all the text. Why did you choose this BPA concentration?

Lines 181: Please, correct the chemical formula (subscript). Correct also a formula in line 270.

I have noticed that no statistical analyses have been performed. Is it correct? Please, clarify and integrate; moreover, indicate how data are shown (means, standard deviations) and how many replicates you have performed.

References

Lines 245, 251 and 287: Please, set the names of genera and species in italics: Pseudomonas putida, Bacillus, and Sphingobium.

Author Response

We have met almost all suggestions of Reviewer 1. For example we have added the information that the biodegradation products are less toxic than bisphenol A, we have explained why we have used the BPA concentration 10 mg/L, etc. Some of the suggestions of Reviewere1 were the same as suggestions of the other Reviewers (e.g. what we mean by pure water). We are grateful for the comments, since they allowed us to significantly improve the manuscript.

The only two suggestions which we have not met are: to clarify in the Materials and Methods section that we refer to the data of chemical analysis we performed (if we understand correctly, since we think that it is clear that in this section we refer to the data which we obtained in this work) and to add in the discussion the possible microbial candidates which could be responsible of the formation of the products observed (as mentioned in the conclusion without biological studies it would speculative).

Reviewer 1 have also asked about statistical analysis. It has been already written in the initial submission (Material and Methods): “Each sample was analyzed three times and the calculated relative standard deviations for the peak areas obtained upon HPLC/UV analysis did not exceed 5%.”.

Reviewer 2 Report

A review of the communication manuscript titled ‟2,2-Bis(4-hydroxyphenyl)-1-propanol - a persistent product of Bisphenol A bio-oxidation in environmental water, as identified by HPLC/UV/ESI-MS” by Drzewiecka et al.

The manuscript is an interesting and valuable contribution to the knowledge of environmental fate of bisphenol A (BPA).

Nevertheless, some corrections, improvements are needed to make it more clear to the readers.

The most evident is that it should be emphasised that BPA was added by spiking to the water samples and that the determination of products was qualitative.

The title ‟2,2-Bis(4-hydroxyphenyl)-1-propanol - a persistent product of Bisphenol A bio-oxidation in environmental water, as identified by HPLC/UV/ESI-MS” should be modified to

‟2,2-Bis(4-hydroxyphenyl)-1-propanol - a persistent product of Bisphenol A bio-oxidation in fortified environmental water, as identified by HPLC/UV/ESI-MS”

Abstract

The sentence ‟ Biodegradation of Bisphenol A in the environmental waters (lake, river and sea) has been studied.” should be prolonged to ‟ Biodegradation of Bisphenol A (BPA) in the environmental waters (lake, river and sea) has been studied on the base of fortification of the samples taken.”

Key Contribution

‟It has been proved, for the first time, that 2,2-bis(4-hydroxyphenyl)-1-propanol can be regarded as persistent product of bisphenol A biodegradation in the environmental waters.” should be modified to ‟It has been proved, for the first time, that 2,2-bis(4-hydroxyphenyl)-1-propanol can be regarded as persistent product of bisphenol A biodegradation in the fortified environmental waters.”

Pg. 1, line 40

‟by dissolution and storage of BPA in water,« should be replaced by ‟by fortification with BPA in water,”

Pg. 2, lines 43 and 44

bisphenol A should be replaced by BPA

Pg. 2, line 52

para-hydroxyacetophenone should be presented also by a chemical fomula

Pg. 2, line 53

what would ‟mineralization” mean?

Pg. 2, lines 61-62

‟Therefore, BPA has been also stored in pure water.” should be replaced by ‟Therefore, BPA has been also added to pure water.” Please define the term ‟pure water”.

Pg. 2, line 63

‟When BPA was stored in…” should be replaced by ‟When BPA was added to…”

Pg. 2, line 65

‟stored in” should be replaced by ‟added to”

Pg. 2, line 66

To my understanding ‟dozen days” should be replaced by ‟days”

Pg. 2, line 71

‟the environmental water samples” should be modified as ‟the fortified environmental water samples”

------------------------------------------------------------------------------------------

Captions to figures should be more descriptive, for example ‟Fig. 2 Exemplary breakdown plots of chromatographic peak areas (UV chromatogram at 280 nm) against days of biodegradation test (BPA, Product 1): (a) Niepruszewo lake, sum- mer; (b) Niepruszewo lake, spring; (c) Warta river, spring.” should be modified to ‟Fig. 2 Exemplary breakdown plots of chromatographic peak areas (UV chromatogram at 280 nm) against days of biodegradation test (BPA, added to water samples at 10 mg/dm3 on day 0, Product 1): (a) Niepruszewo lake, sum- mer; (b) Niepruszewo lake, spring; (c) Warta river, spring.”

---------------------------------------------------------------------------------------------------------

Pg. 3, line 88

‟at the retention time 12.6” should be replaced by ‟at the retention time of 12.6 min”

Pg. 4

What was analysed in Fig. 3?

Pg. 4, line 106

What does CID mean?

Pg. 4

It is necessary to present a schematic presentation of chemical process evaluated with formulas.

Was ratio between the products different between fresh and marine water samples?

Pg. 5, line 131

The source of p-HAP standard should be reported in the Materials and Methods

Conclusions

Did the authors observe any pattern of BPA degradation e.g. influence of water source or time of the year?

Pg. 6, line 160

Bisphenol A was obtained from Sigma-Aldrich (Poznań, Poland): purity needs to be reported.

Pg. 6, lines 161-163

Geographical position of the water sources should be defined.

Pg. 6, line 182

CH3CN-H2O

Pg. 6, line 186

Ratios should be defined for example as (v/v).

Pg. 6, line 207

Q-TOF should be reported in brief (instrument, producer).

References

DOI should be reported if available.

Supplementary material

Figure S1

Units should be in brackets

‟Figure S1. Linearity obtained during HPLC/UV and HPLC/MS analysis of BPA solutions (peak area is in the arbitrary units).” should be modified as ‟Figure S1. Linearity obtained during HPLC/UV and HPLC/MS analysis of BPA water solutions (peak area is in the arbitrary units).”

Figure S2

‟Figure S2. Exemplary UV chromatograms obtained at 230 nm (acidified gradient) for BPA stored at sea pure water; the biodegradation process has not been observed.” should be modified as ‟Figure S2. Exemplary UV chromatograms obtained at 230 nm (acidified gradient) for BPA added to sea pure water; the biodegradation process has not been observed.” The term ‟sea pure water” needs to be defined.

Figure S3

‟Figure S3. Exemplary UV chromatograms obtained at 230 nm (acidified gradient) for BPA stored at sea water; very slow biodegradation process has been observed.” should be

modified as ‟ Figure S3. Exemplary UV chromatograms obtained at 230 nm (acidified gradient) for BPA added to sea water; very slow biodegradation process has been observed”.

Author Response

We have met most of the suggestions of Reviewer 2, some of them were the same as suggestions of the other Reviewers, e.g. to explain in more details the origin of environmental water used in the experiments. We are grateful for the comments, since they allowed us to significantly improve the manuscript. The following suggestions have not been met or have met in the initial submission:

  1. Reviewer 2 has suggested to remove “dozen” in the sentence “When BPA was stored in the sea water its concentration was only slightly lowered after several dozen days, no biodegradation products were detected (supplementary material, Figure S3). We think that the use “dozen” better express the meaning of the sentence.
  2. Reviewer 2 has suggested to present a schematic presentation of chemical process evaluated with formulas. We think that, on the basis of data obtained, such presentation would speculative.
  3. Reviewer 2 has asked “Was ratio between the products different between fresh and marine water samples?” We do not understand this question.
  4. Reviewer 2 has asked “Did the authors observe any pattern of BPA degradation e.g. influence of water source or time of the year”. It has been already written in the initial submission (different lakes and seasons).
  5. Reviewer 2 has written “Pg. 6, line 186 Ratios should be defined for example as (v/v).” We do not understand this suggestion, since we think that the gradient used is described correctly.
  6. Reviewer 2 has written “Q-TOF should be reported in brief (instrument, producer)”. It has been already written in the initial submission, in the supplementary material.
  7. Reviewer 2 has suggested to add DOI to the references. We think that the reference style is in agreement with the journal requirements.

Reviewer 3 Report

The study was well performed and after the suggested additions and changes it will be suitable for the journal. Language polishing by a native English speaker is highly recommended.

Author Response

Reviewer 3 has marked the suggestions on the pdf file provided. We have met all the suggestions (some of them were the same as suggestions of the other Reviewers).